# Epidemiological and Immunological Features of Obesity and SARS-CoV-2

**DOI:** 10.3390/v13112235

**Published:** 2021-11-06

**Authors:** Eric J. Nilles, Sameed M. Siddiqui, Stephanie Fischinger, Yannic C. Bartsch, Michael de St. Aubin, Guohai Zhou, Matthew J. Gluck, Samuel Berger, Justin Rhee, Eric Petersen, Benjamin Mormann, Michael Loesche, Yiyuan Hu, Zhilin Chen, Jingyou Yu, Makda Gebre, Caroline Atyeo, Matthew J. Gorman, Alex Lee Zhu, John Burke, Matthew Slein, Mohammad A. Hasdianda, Guruprasad Jambaulikar, Edward W. Boyer, Pardis C. Sabeti, Dan H. Barouch, Boris Julg, Adam J. Kucharski, Elon R. Musk, Douglas A. Lauffenburger, Galit Alter, Anil S. Menon

**Affiliations:** 1Brigham and Women’s Hospital, Boston, MA 02115, USA; gzhou5@bwh.harvard.edu (G.Z.); bmormann@mgh.harvard.edu (B.M.); michael.loesche@mgh.harvard.edu (M.L.); mhasdianda@bwh.harvard.edu (M.A.H.); gjambaulikar@bwh.harvard.edu (G.J.); eboyer@bwh.harvard.edu (E.W.B.); 2Harvard Medical School, Boston, MA 02115, USA; 3Harvard Humanitarian Initiative, Boston, MA 02114, USA; mdestaubin@hsph.harvard.edu; 4Massachusetts Consortium on Pathogen Readiness, Boston, MA 02115, USA; pardis@broadinstitute.org (P.C.S.); galter@mgh.harvard.edu (G.A.); 5Computational and Systems Biology Program, Massachusetts Institute of Technology, Cambridge, MA 02139, USA; sameed@mit.edu; 6Broad Institute of MIT and Harvard, Cambridge, MA 02142, USA; dbarouch@bidmc.harvard.edu (D.H.B.); bjulg@mgh.harvard.edu (B.J.); 7Ragon Institute of MGH, MIT and Harvard, Cambridge, MA 02139, USA; sfischinger@mgh.harvard.edu (S.F.); ybartsch@mgh.harvard.edu (Y.C.B.); zchen30@mgh.harvard.edu (Z.C.); jyu36@mgh.harvard.edu (J.Y.); mgebre@bidmc.harvard.edu (M.G.); catyeo@mgh.harvard.edu (C.A.); mGorman8@mgh.harvard.edu (M.J.G.); ALZHU@mgh.harvard.edu (A.L.Z.); JSBURKE@mgh.harvard.edu (J.B.); matthew.d.slein.gr@dartmouth.edu (M.S.); 8Space Exploration Technologies Corp., Hawthorne, CA 90250, USA; mgluck7@gmail.com (M.J.G.); sbeger@email.arizona.edu (S.B.); justin_rhee@brown.edu (J.R.); eric.petersen@ucdenver.edu (E.P.); yiyuan.hu.med@dartmouth.edu (Y.H.); jehn@spacex.com (E.R.M.); anil.menon@spacex.com (A.S.M.); 9Harvard T.H. Chan School of Public Health, Boston, MA 02115, USA; 10Howard Hughes Medical Institute, Chevy Chase, MD 20815, USA; 11Center for Virology and Vaccine Research, Beth Israel Deaconess Medical Center, Boston, MA 02115, USA; 12London School of Hygiene and Tropical Medicine, London WC1E 7HT, UK; Adam.Kucharski@lshtm.ac.uk; 13Department of Biological Engineering, Massachusetts Institute of Technology, Cambridge, MA 02139, USA; lauffen@mit.edu

**Keywords:** SARS-CoV-2, COVID-19, body mass index, obesity, epidemiology, clinical features, immunity

## Abstract

Obesity is a key correlate of severe SARS-CoV-2 outcomes while the role of obesity on risk of SARS-CoV-2 infection, symptom phenotype, and immune response remain poorly defined. We examined data from a prospective SARS-CoV-2 cohort study to address these questions. Serostatus, body mass index, demographics, comorbidities, and prior COVID-19 compatible symptoms were assessed at baseline and serostatus and symptoms monthly thereafter. SARS-CoV-2 immunoassays included an IgG ELISA targeting the spike RBD, multiarray Luminex targeting 20 viral antigens, pseudovirus neutralization, and T cell ELISPOT assays. Our results from a large prospective SARS-CoV-2 cohort study indicate symptom phenotype is strongly influenced by obesity among younger but not older age groups; we did not identify evidence to suggest obese individuals are at higher risk of SARS-CoV-2 infection; and remarkably homogenous immune activity across BMI categories suggests immune protection across these groups may be similar.

## 1. Introduction

Obesity is a key risk factor for severe disease and death from novel coronavirus disease 2019 (COVID-19) [1,2], the disease caused by severe acute respiratory syndrome coronavirus 2 (SARS-CoV-2). With over 1.9 billion people overweight or obese globally [3], implications for SARS-CoV-2 morbidity and mortality are substantial. After adjusting for age and obesity-related comorbidities such as diabetes, hypertension, and coronary heart disease, obesity remains a strong independent predictor of excess morbidity and mortality [1,4,5]. These findings are not entirely unexpected [6]. Obesity and poor clinical outcomes have been described with other viral pathogens, most notably influenza A (H1N1) during the 2009 pandemic, when obesity was associated with increased hospitalizations, need for intensive care support, and deaths [7,8]. In addition to the relationship between obesity and clinical outcomes, emerging evidence suggests a link between higher body mass index (BMI) and higher incidence rates of COVID-19 or SARS-CoV-2 infection [4,9,10], suggesting increased BMI may enhance susceptibility to infection, with important implications for individual-level risks and population-level transmission dynamics [4]. The role of obesity on the immune response to SARS-CoV-2 has also been the focus of intense attention [11]. Obesity has been linked to less robust and/or effective immune response after natural influenza infection [12] or vaccination [13], raising concerns about diminished protective immunity following natural SARS-CoV-2 infection or vaccination [4,14]. Yet, results from SARS-CoV-2 vaccine trials suggest similar levels of protection across BMI categories [15]. Thus, given substantial uncertainly about multiple features of obesity and SARS-CoV-2, we investigated if BMI is associated with differential (i) risks of testing positive for anti-SARS-CoV-2 IgG antibodies, (ii) symptom phenotype, and (iii) adaptive immune features.

## 2. Materials and Methods

### 2.1. Study Design, Setting and Study Population

This study examines data from a prospective observational cohort study using serial serological assessment to characterize the immunoepidemiology of SARS-CoV-2 infection among industry employees. Serostatus was unknown at the time of subject enrollment. The study population was comprised of Space Exploration Technologies Corporation employees, all of whom were invited to participate. Study enrollment commenced 20 April and employees were invited to participate on a rolling basis through 28 July 2020; 4469 volunteered and were enrolled from ~8400 total employees from seven work locations in four US states. Serial blood sampling and interim symptom reporting were performed monthly.

### 2.2. Covariates

Standardized data measures included demographic and medical history variables (listed in Table 1) and COVID-19 compatible symptoms between 1 March 2020, and study enrollment. Symptoms were classified as primary (fever, chills or feverish, cough, anosmia, ageusia) and other compatible symptoms (body or muscle aches, sore throat, nausea or vomiting, diarrhea, congestion, and increased fatigue/generalized weakness). Blood was sampled and interim symptoms were monitored monthly.

### 2.3. Laboratory Analyses

Serological analyses were performed using the Ragon/MGH enzyme-linked immunosorbent assay, which detects IgG against the receptor binding domain (RBD) of the SARS-CoV-2 spike glycoprotein using a previously described method [16] (Appendix B). Assay performance has been externally validated in a blinded fashion at 99.6% specific and benchmarked against commercial EUA approved assays [17]. Immune profiling methods are detailed in Annex 1. Briefly, specific antibody subclasses and isotypes and FcγR binding against SARS-CoV-2 RBD, nucleocapsid and full spike proteins were assessed using a custom Luminex multiplexed assay (Luminex Corp, Austin, TX, USA). Viral neutralization was assessed on a SARS-CoV-2 pseudovirus assay, as described previously [18] with neutralization titer defined as the sample dilution associated with a 50% reduction in luminescent units. The presence of neutralizing activity was defined as a titer >20. T cell activity was assessed on an enzyme-linked interferon-gamma immunospot assay with >25 spot forming cells per 10^6^ peripheral blood mononuclear cells considered positive.

### 2.4. Data Classification and Analyses

Seropositivity was determined by the detection of SARS-CoV-2 specific IgG. BMI was calculated by dividing weight in kilograms by height in meters squared and categorized by underweight (<18.5 kg/m^2^), normal weight (18.5 to 24 kg/m^2^; reference), overweight (25 to 29 kg/m^2^), obesity class 1 (30 to 34 kg/m^2^), obesity class 2 (35 to 39 kg/m^2^), and obesity class 3 or severe obesity (≥40 kg/m^2^) according to the World Health Organization and the US Centers for Disease Control and Prevention.

We performed discrete analyses to address the three aims of the study. For assessment of risk of seropositivity by BMI, the primary exposure of interest was BMI and the outcome variable of interest was seropositivity at any time point. We assessed the unadjusted association between a range of demographic (*n* = 7) and medical history (*n* = 17) covariates using χ^2^ to compare proportions and ANOVA or Kruskal–Wallis tests to compare means. For adjusted analyses, we constructed a multivariable logistic regression model that included, in addition to BMI and serostatus, age, sex, ethnicity, race, comorbidities, primary work location, number of individuals in the household, and children in the household; variables with a *p*-value <0.10 were assessed by backward elimination and excluded if the *p*-value was >0.10 and did not meaningfully alter the point estimates of the remaining variables. The risk of being seropositive is expressed as odds ratios (ORs) with binomial exact 95% confidence intervals (CIs). *p*-values <0.05 were considered statistically significant. To understand if obesity status is associated with differential reporting of symptoms, we computed the proportion of seropositive individuals reporting COVID-19 compatible symptoms stratified by obesity status. Symptoms were analyzed from the period preceding the first seropositive result. For example, if an individual was seronegative at baseline and seropositive at the subsequent time point, the symptoms reported between those timepoints were analyzed. The primary exposure of interest was obesity, and symptoms the outcome variables of interest. Given data suggesting the adverse impact of obesity on COVID-19 mortality may decline with age [9], we assessed if similar age-dependent obesity risk may be observed for symptom reporting by conducting subgroup analysis stratified by < or ≥40 years, with categorization selected due to sparsity of older participants. Lastly, given an accumulation of evidence that obesity impairs the immune response to a range of pathogens [6,13,19,20,21], we stratified 20 discrete immune features by obesity status to identify univariate differences.

We additionally performed uniform manifold approximation and projection (UMAP) [22] a mathematical approach for exploratory analyses that constructs a visualizable summary of multiple subjects’ characteristics, with each point representing an individual and clusters representing underlying uniformities in subject characteristics. Luminex UMAP and Mann–Whitney U Tests were conducted using scikit-learn, a machine learning toolkit for the Python programming language. Analyses were performed using the R software package (Version 4.0, www.R-project.org/, accessed 1 July 2020) or the Python programming language (Version 3.7, python.org). All available relevant data was included.

## 3. Results

A total of 4469 study participants from ~8400 total employees (53%) were enrolled. Baseline characteristics are listed in Table 1. Mean BMI was 27.1 kg/m^2^ (SD 5.4) with a median of 25.8 kg/m^2^ (range 15.6–60.9). Most subjects were normal weight (18.5–24 kg/m^2^) (1686, 39.5%) or overweight (25–29 kg/m^2^) (1523, 35.7%), and 24.1% and 0.80% met criteria for obese (≥30 kg/m^2^) and underweight (≤18.5 kg/m^2^), respectively. A total of 322 out of 4469 (7.21%) study participants were seropositive, of which five (1.6%) were hospitalized and none required critical care support or died. Unadjusted rates are detailed in Table 1 and were higher in South Texas (OR 4.28 [95% CI, 2.54 to 7.21), *p* < 0.0001]) and among Hispanics (2.91 [95% CI 2.26–3.75], *p* < 0.0001); and were lower in Seattle, Washington (0.30 [95% CI, 0.11 to 0.82], *p* = 0.02]), and in the 30–39 year age group (0.72 [95% CI, 0.56 to 0.94], *p* = 0.02). Multivariable regression analyses retained only primary work location as significantly associated with serostatus, with increased risk in South Texas (OR 4.28 [2.54–7.21], *p* < 0.0001) and lower risk in Seattle (OR 0.30 [0.11–0.82], *p* = 0.02), largely reflecting local transmission rates. The strong univariate association between Hispanic ethnicity and serostatus was not retained after adjusting for work location (OR 1.27 [0.94–1.73], *p* = 0.12).

### 3.1. BMI and Serostatus

A total of 4270 out of 4469 participants (95.5%) provided weight and height data and are included in BMI analyses. Unadjusted risks of seropositivity stratified by BMI are listed in the Table 1; only BMI 30 to 34 kg/m^2^ (versus normal/healthy weight, 18.5–24 kg/m^2^) was associated with differential serostatus (OR 1.48 [1.06 to 2.05], *p* < 0.02). However, after adjusting for all candidate variables (Table 1), no association was detected. Rather, higher BMI and in particular severe obesity (BMI ≥40 kg/m^2^) trended non-significantly to lower seroprevalence (Figure 1A). Subgroup analysis from a single high prevalence location where, given the high force of infection as evidenced by high seroprevalence (22.5% versus 4.2% for all other sites combined), we predict risks for infection, including any effect of BMI, would be more clearly delineated (Appendix A). Findings were similar to the primary analysis with no evidence of increased seroprevalence with increasing BMI and point prevalence measures consistently trended lower than normal/healthy weight (Figure 1B).

### 3.2. BMI and COVID-19 Compatible Symptoms

Of 262 seropositive participants with complete symptom data, three (1.1%) were underweight (<18.5 kg/m^2^), 89 (34.0%) normal weight (18.5–24 kg/m^2^), 89 (34.0%) overweight (25–29 kg/m^2^), and 81 (30.9%) obese (≥30 kg/m^2^). A total of 106/262 (40.5%) reported one or more of 11 COVID-19- compatible symptoms and 68/262 (26.0%) reported one or more of five primary COVID-19 symptom. When comparing symptoms between normal weight and overweight (but not obese) individuals, there were no meaningful differences or trends (Appendix A) and, therefore, subsequent analyses were stratified by obese versus non-obese. Obesity was associated with increased reporting of multiple symptoms including fever, chills or feverish but no measured fever, myalgias, and ≥6 symptoms (Figure 2). Except for congestion (OR 0.87 [0.43–1.70]), a similar and consistent but non-significant trend was observed for all symptoms. Overall, obese individuals registered more symptoms and more primary symptoms. Age appears to play an important role when assessing obesity and symptom phenotype and fever was more commonly reported among obese vs. non-obese individuals under 40 years of age (OR 4.99 [1.97–13.35]) but not over 40 years (OR 1.32 [0.30–5.57]). Similarly, reporting ≥6 symptoms was more common among obese vs. non—obese under 40 years (OR 3.0 [1.32–6.85]) but not for those greater than 40 years (OR 0.94 [0.18–4.26]). A strikingly similar trend was observed for most other symptoms and aggregate symptom measures (Figure 3). To understand if similar age-dependent effects may be observed among younger age groups, we performed subgroup analyses on 19–29 versus 30–39 year age groups; no similar age-dependent effects were observed (Figure 4).

### 3.3. Obesity and Functional Immune Response

Among the same 262 seropositive individuals, peak SARS-CoV-2 RBD IgG titers were 0.92 ug/mL (SD 2.47) among obese (*n* = 81) and 1.12 ug/mL (SD 3.21) among non-obese (*n* = 181) participants (*p* = 0.601). Deep immune profiling was performed among a subset of 77 participants including 25 obese and 52 non-obese individuals. Mean ELISA NC IgG titers were 0.35 (SD 0.48) among obese versus 0.30 (0.34) among non-obese individuals (*p* = 0.57). Viral neutralization activity was detected in 3/25 (12.0%) and 6/52 (11.5%) of obese and non-obese individuals, respectively (*p* = 0.95). When assessing 20 immune features measured by Luminex, no univariate differences were observed between obesity categories, with sparse levels across both obese and non-obese individuals tightly linked to antibody titers (Figure 5A, Appendix A). Similarly, no clustering or trends between BMI and immunological features were identifiable either by UMAP (Figure 5B) or Spearman’s correlation (Figure 5C). Lastly, given evidence that T cells may be key mediators of adaptive immunity in SARS-CoV-2, we examined responses to nucleocapsids protein or spike protein overlapping peptide pools quantified by IFN-g ELISpot among 12 obese and 28 non-obese individuals. There was no difference in the proportion with SARS-CoV-2 T cell activity (≥25 SFC/10^6^ PBMCs) against nucleocapsid peptides (3/12 [25%] versus 7/28 [25.0%]) or spike peptides (3/12 [25%] versus 7/28 [25.0%]). In fact, the only difference observed was higher SFC against nucleocapsid (mean 124 SFC/10^6^ PBMCs versus 47 SFC/10^6^ PBMCs, *p* = 0.02), but not spike (44 SFC/10^6^ PBMCs versus 44 SFC/10^6^ PBMCs, *p* = 1.00), among obese versus non-obese individuals with T cell activity.

## 4. Discussion

We present data from a multi-site prospective cohort of non-hospitalized individuals unbiased to serostatus at study entry to investigate the association between BMI, SARS-CoV-2 serostatus, symptom phenotype, and functional and non-functional immune measures. Given the prevalence of overweight/obese among adults is close to 70% in most high-income countries and ≥50%, in many lower- and middle-income countries, the scientific and public health implications for the current pandemic are substantial [4]. By combining traditional epidemiological approaches with deep immune profiling, these data provide key insights into the epidemiology and immune characteristics of obesity in SARS-CoV-2 infections.

Studies that report an increased risk of COVID-19 or SARS-CoV-2 infection with higher BMI are intriguing and raise essential questions about factors driving transmission. Given the global burden of obesity, delineating risks for infection is a public health priority. Interestingly, our findings diverge from published reports that examine the risk for COVID-19 by BMI, including a nationwide case-control study from South Korea [9] and a cross-sectional study from a primary care surveillance network in the United Kingdom [23], that identified an increased risk of COVID-19 with increasing BMI. A meta-analysis of 20 studies reported a pooled increased risk of 46.0% (OR = 1.46; 95% CI, 1.30–1.65; *p* < 0.0001) with 18 of 20 studies demonstrating higher COVID-19 risk among obese individuals [4]. Notably, our study did not identify an increase in adjusted seroprevalence with increasing BMI and, conversely, identified a trend to lower infection risk with higher obesity classes. This trend was consistent when both considering all data and when performing subgroup analysis on a high transmission site where the increased force of infection may more precisely delineate heterogeneity in infection risks. Reasons for the difference between our and prior study outcomes are likely multifactorial, with differences in study design, obesity classification, and population behaviors likely influencing findings. However, a key difference is we examined SARS-CoV-2 infection risk using serological methods unbiased to exposure risks or presence or absence of symptoms at study entry versus prior studies that examined risks for clinically apparent infection (i.e., COVID-19). As such, our primary outcome measure was SARS-CoV-2 infection rather than clinically apparent disease, a key difference that likely contributes to differences in study findings and conclusions. Given individuals with obesity are more likely to experience fever and multiple other symptoms with SARS-CoV-2 infection, as our data indicates, this population is more likely to be tested and over-represented in studies that identify study participants through routine surveillance approaches [24,25].

Our finding that obesity is associated with increased COVID-19 compatible symptoms among SARS-CoV-2 seropositive individuals provides benchmark data for understanding symptom heterogeneity in mild infections by BMI. We demonstrate that not only are well established measures of severe disease such as hospitalization, intensive care requirements and death more common among obese individuals [4,5], but obesity is also an important driver of fever and other symptomatology in non-severe infections. These findings may be due to a dysregulated inflammatory response which is characteristically associated with obesity, that when exposed to a secondary pro-inflammatory inflammatory stimulus, such as during SARS-CoV-2 infection, leads to augmented circulation of pro-inflammatory cytokines. While our data does not define the mechanism driving these findings, it informs our understanding of symptom phenotype and obesity, guides our interpretation of epidemiological data, and highlights potential implications of using passively collected symptom-driven surveillance data to characterize the epidemiology of infectious pathogens. We also identify an intriguing influence of age on obesity and symptom phenotype, with a compelling association below 40 years of age but near complete absence of effect in older adults. These findings are notable given they imply the established interaction between obesity and age on COVID-19 morbidity and mortality, with obesity disproportionately driving increased disease severity among younger age groups [5,26], extend throughout the spectrum of disease and are not restricted to severe disease.

Given the fundamental role of the adaptive immune response in both the resolution of infection and the severity of disease [27], we also probed multiple binding and functional immune markers to assess differential immune responses by obesity status. While previous studies noted poor seroconversion and inadequate seroprotection across vaccine trials targeting other pathogens [28], we did not detect meaningful differences in binding or neutralizing antibodies, T cell activity, or other functional humoral measures by BMI among SARS-CoV-2 infections. These findings, while notable, should be considered in the context of this cohort in which >98% of infections were asymptomatic or mild and, therefore, may not capture the full range of disease burden associated with SARS-CoV-2. Yet, the overlapping and indistinguishable antibody and T-cell helper profiles point to unaltered adaptive immunity with BMI, raising the possibility that BMI-driven immunological changes during SARS-CoV-2 infection may manifest largely within the innate immune response. Significant alterations in chronic inflammation, particularly driven by persistent innate cytokine responses from adipocytes including IL-6, TNF-α, Type 1 IFN, and Leptin (Figure 6, modified from Alarcon, 2021), have been noted in the setting of obesity [29]. Dissecting the influence of adipocyte inflammatory responses, associated cytokine storm, and enhanced symptomatology, particularly among individuals with a high BMI, may point to mechanistic differences in viral sensing across populations. These data point to remaining knowledge gaps on the relative importance and interplay of the humoral, cellular, and innate immunity in SARS-CoV-2 infection and disease.

Although this study is unique in combining a large prospective, multisite serology-based SARS-CoV-2 cohort with deep immune profiling, there are limitations. The study population are industry employees with higher representation of Hispanic ethnicity, white race, male sex, and younger individuals with less comorbidities than the US population; therefore, findings may not be generalizable. Our study enrolled around 50% of the eligible population, which introduces the potential for ascertainment bias, which may again impact the generalizability of our findings. Given antibody decay can lead to seroreversion (from seropositive to seronegative), some seronegative study participants may have been previously infected. Seroreversion would be expected to occur more frequently among individuals that generated only a weak immune response but given we do not observe a systematic difference between obese and non-obese individuals across multiple immune parameters, including peak anti-RBD IgG titers, we think this is unlikely to meaningfully impact our findings. Other potential study limitations should be noted but their impact would be expected to be evenly distributed across cohort participants and therefore not introduce a systematic bias and impact study findings. These include (i) limited recall of COVID-19 compatible symptoms; (ii) delayed seroconversion relative to reported symptoms, so depending on timing of infection and blood sampling, some registered symptoms may not be due to SARS-CoV-2 infection; and (iii) false positive serological screening results. Lastly, behavioral factors, which can be critical drivers of transmission and may be associated with BMI, were not assessed in this study.

## 5. Conclusions

We demonstrate that obesity influences symptom phenotype in mild COVID-19 infections, suggesting obesity impacts the pathophysiology of COVID-19 throughout the spectrum of disease severity. Our findings do not, however, suggest that obesity increases susceptibility to SARS-CoV-2 infection. Nor did we identify immunological features differentiating obese from non-obese individuals across mild and asymptomatic infection, a hopeful signal that both natural infection- and vaccine-induced protective immunity may be similar across these populations.

## Figures and Tables

**Figure 1 viruses-13-02235-f001:**
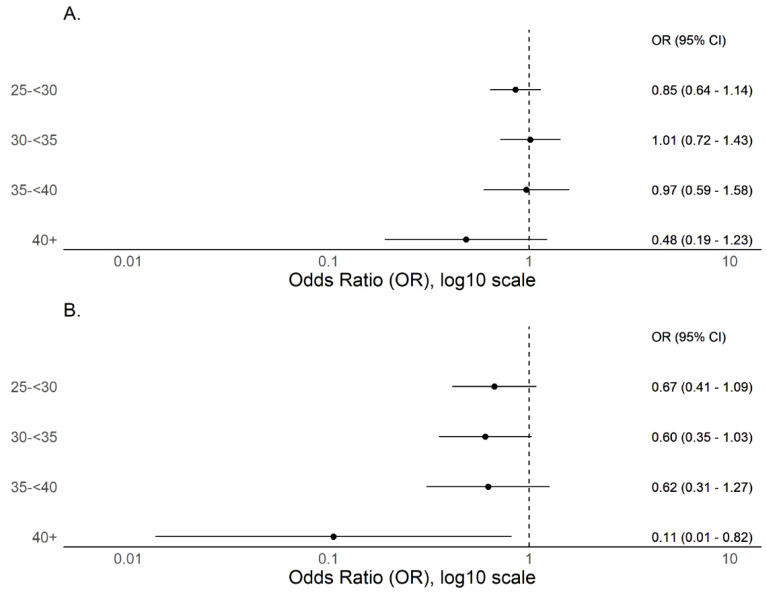
Forest plots of adjusted odds ratio for seropositivity by BMI as a categorical variable with normal BMI (18.5–<25) as reference. (**A**) Includes participants with BMI measures and demonstrates a non-significant trend to declining seroprevalence with BMI ≥40 kg/m^2^ when compared to normal/healthy weight (BMI 18.5–24 kg/m^2^) (*n* = 4270). (**B**) Includes only participants from a single high seroprevalence (22.5%) location in South Texas, where the high force of infection may more clearly delineate infection risks (*n* = 629).

**Figure 2 viruses-13-02235-f002:**
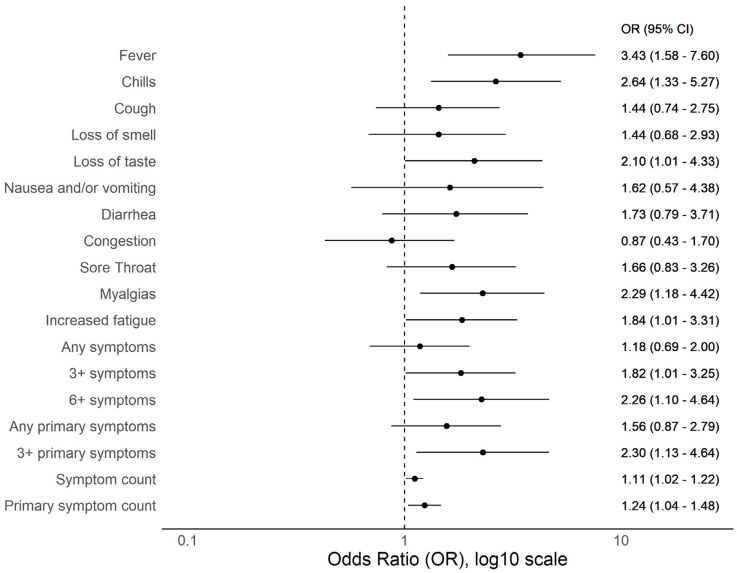
Forest plot of odds ratios of reported COVID-19 compatible symptoms among obese (*n* = 85) versus non-obese (*n* = 179) SARS-CoV-2 seropositive individuals.

**Figure 3 viruses-13-02235-f003:**
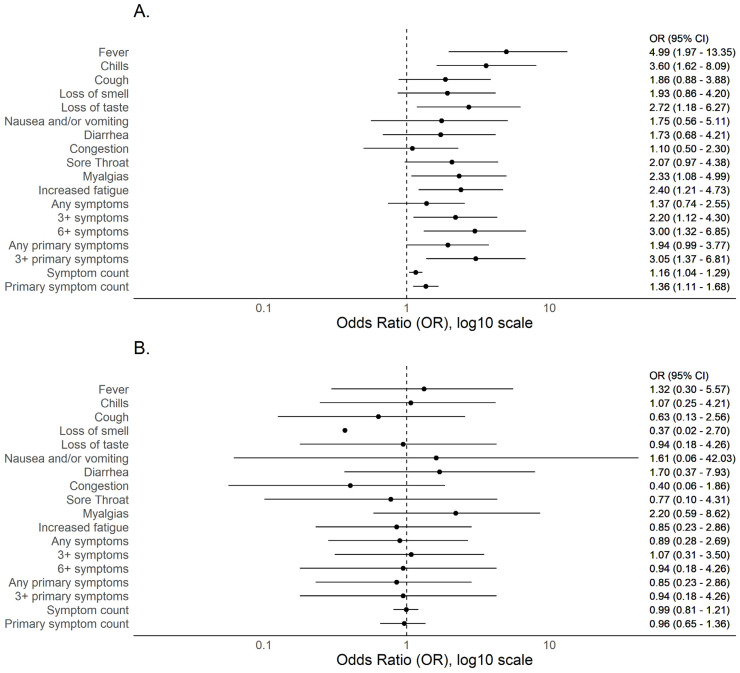
Forest plot of odds ratios of COVID-19 compatible symptoms among obese versus non-obese SARS-CoV-2 seropositive individuals stratified by (**A**) <40 years (*n* = 195) and (**B**) ≥40 years (*n* = 67).

**Figure 4 viruses-13-02235-f004:**
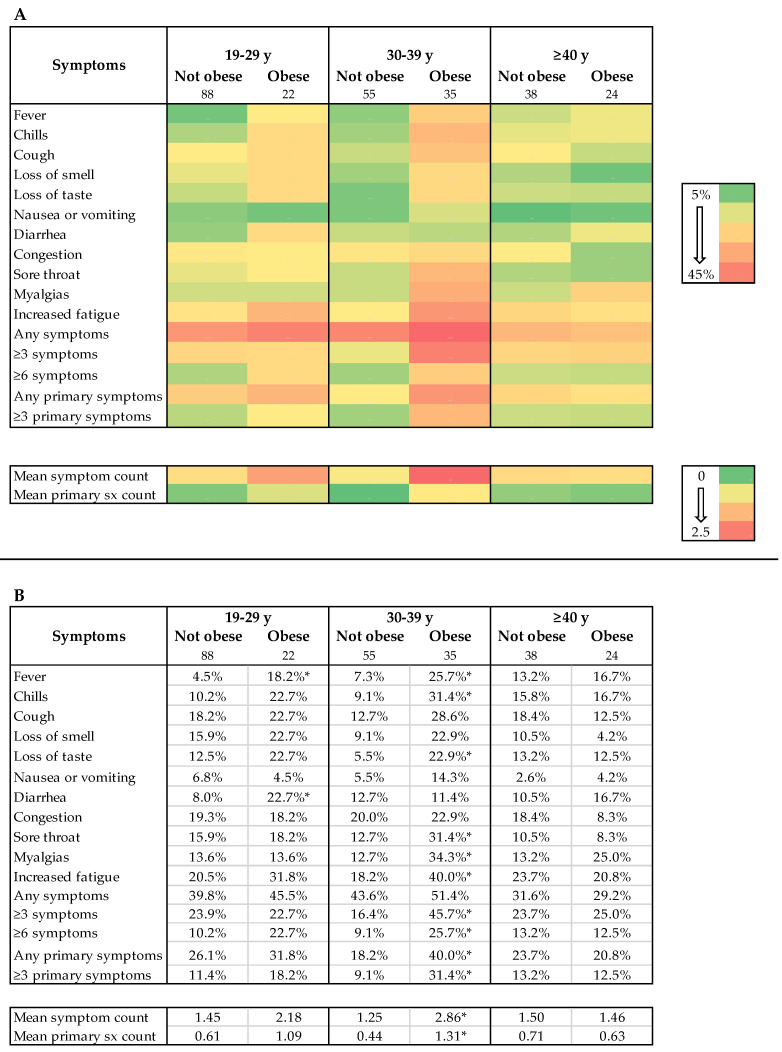
Symptom reporting by age group and obesity status among SARS-CoV-2 seropositive individuals. (**A**) Heatmap shows consistently higher symptom reporting amongst obese individuals in the 19–29 and 29–39 year age groups but not ≥40-year age group. Number of individuals in each category are listed below obesity markers. (**B**) Table lists relevant values. * indicates *p* < 0.05 for difference between obese and non-obese in that age category with Chi-squared test for proportions and ANOVA for test of mean.

**Figure 5 viruses-13-02235-f005:**
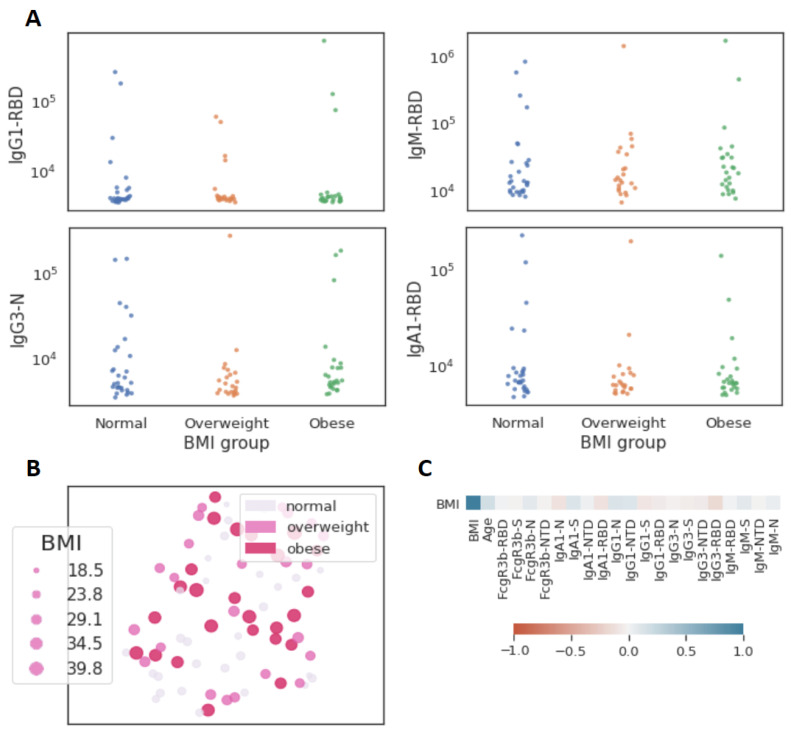
Limited influence of BMI on SARS-CoV-2 antibody profiles (*n* = 77). (**A**) The dot plots show similar mean fluorescent intensity levels of IgG1, IgM, IgG3, and IgA levels across individuals classified as normal weight (*n* = 29), overweight (*n* = 23), and obese (*n* = 25). (**B**) The uniform manifold approximation and projection (UMAP) shows the relationship between antibody profiles and BMI (dot size, color intensity), highlighting the limited influence of BMI on shaping SARS-CoV-2 antibody responses. (**C**) Correlation plot of shows limited correlation between BMI and 20 immunological features.

**Figure 6 viruses-13-02235-f006:**
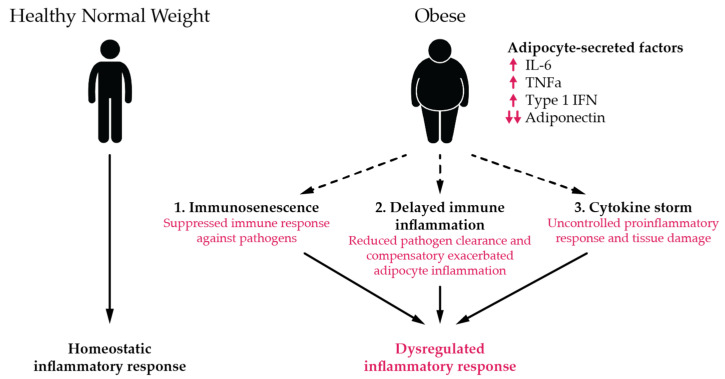
Role of obesity in inflammatory response to infection. Adipocyte-secreted factors (e.g., adiponectin, leptin, Type I IFNs, and IL-6) contribute to normal homeostatic immune responses against infectious pathogens among healthy/normal weight. Obesity-dependent changes in adipocyte function can contribute to (1) immunosenescence (suppressed immune response against pathogens); (2) delayed immune inflammation (reduced pathogen clearance and compensatory exacerbated adipocyte inflammation); and (3) “cytokine storm” (IL-). Modified from Alarcon, 2021 [29].

**Table 1 viruses-13-02235-t001:** Characteristics, serostatus, and unadjusted odds ratios of study participants.

Covariate ^1^	All Participants (*n* = 4469)	Seropositive Participants (*n* = 322)		OR (95% CI)	*p*-Value ^5^
	N	N	%		
**Age group**					
18–29 y	1668	133	8.0%	ref	
30–39 y	1761	104	5.9%	0.72 (0.56 to 0.94)	0.0174 *
40–49 y	584	50	8.6%	1.08 (0.77 to 1.52)	0.6545
50–59 y	315	26	8.3%	1.04 (0.67 to 1.61)	0.8666
60+ y	85	2	2.4%	0.28 (0.07 to 1.14)	0.076
**BMI**					
<18.5	34	3	8.8%	1.44 (0.43 to 4.80)	0.5500
18.5–<25	1686	106	6.3%	ref	
25–<30	1523	101	6.6%	1.06 (0.80 to 1.40)	0.6916
30–<35	676	61	9.0%	1.48 (1.06 to 2.05)	0.0196 *
35–<40	246	23	9.3%	1.54 (0.96 to 2.47)	0.0742
≥40	105	5	4.8%	0.75 (0.30 to 1.87)	0.5308
**Ethnicity**					
Not Hispanic/Not Latinx	2492	113	4.5%	ref	
Hispanic/Latinx	1274	155	12.2%	2.91 (2.26 to 3.75)	<0.0001 ****
**Race**					
White	2862	185	6.5%	ref	
American Indian/Alaska Native	32	3	9.4%	1.50 (0.45 to 4.96)	0.5092
Asian	442	18	4.1%	0.61 (0.37 to 1.01)	0.0535
Black	72	2	2.8%	0.41 (0.10 to 1.70)	0.2207
Native Hawaiian/Pacific Islander	29	2	6.9%	1.07 (0.25 to 4.54)	0.9249
More than one race	292	13	4.5%	0.67 (0.38 to 1.20)	0.1796
**Sex ^2^**					
Female	600	40	6.7%	ref	
Male	3730	267	7.2%	1.08 (0.77 to 1.52)	0.6634
**Children ≤ 18 y in household**					
No	3014	204	6.8%	ref	
Yes	1342	106	7.9%	1.18 (0.93 to 1.51)	0.1808
**No. in household**					
1	640	41	6.4%	ref	
2–4	3027	214	7.1%	1.11 (0.79 to 1.57)	0.5490
>4	659	51	7.7%	1.23 (0.80 to 1.88)	0.3499
**Primary work location**					
Cape Canaveral, Florida	268	17	6.3%	ref	
Hawthorne, California	2859	111	3.9%	0.60 (0.35 to 1.01)	0.0544
McGregor, Texas	257	21	8.2%	1.31 (0.68 to 2.55)	0.4202
Seattle, Washington	253	5	2.0%	0.30 (0.11 to 0.82)	0.0190 *
South Texas, Texas	712	160	22.5%	4.28 (2.54 to 7.21)	<0.0001 ****
Other	69	1	1.4%	0.23 (0.03 to 1.79)	0.1623
**Comorbidities ^3,4^**					
Asthma	368	20	5.4%	0.72 (0.45 to 1.15)	0.1721
Hypertension	356	26	7.3%	1.02 (0.67 to 1.54)	0.9405
Diabetes mellitus	101	11	10.9%	1.59 (0.84 to 3.01)	0.1509
Coronary heart disease	17	1	5.9%	0.80 (0.11 to 6.08)	0.8329
Stroke	9	2	22.2%	3.70 (0.76 to 17.87)	0.1039
Emphysema/COPD	9	1	11.1%	1.61 (0.20 to 12.93)	0.6532
Cancer—not receiving treatment	39	2	5.1%	0.69 (0.17 to 2.89)	0.6163
Other lung disease	26	2	7.7%	1.07 (0.25 to 4.56)	0.9233
Other immunocompromised	61	4	6.6%	0.92 (0.33 to 2.55)	0.8710
Other chronic medical condition	176	9	5.1%	0.72 (0.36 to 1.43)	0.3471
**Smoking history**					
Never	3769	263	7.0%	ref	
Prior	367	24	6.5%	0.93 (0.61 to 1.44)	0.7514
Current	229	23	10.0%	1.49 (0.95 to 2.33)	0.0826

^1^ Not reported data: age group (*n* = 56), BMI (199), ethnicity (703), race (740), sex (139), children in HH (113), No. in HH (143), primary location (51), comorbidities (105). ^2^ Four (4) reported “other sex”, none were seropositive. ^3^ For comorbidities reference value for OR is no. COPD chronic obstuctive pulmonary disease. ^4^ Other comorbidities with no seropositive participants: chronic kidney disease (10), Heart failure (4), Cancer receiving treatment (3), Other heart disease (22). ^5^ *p*-values unadjusted for multiple hypothesis testing: * <0.05, **** <0.0001.

## Data Availability

The data presented in this study are available on reasonable request from the corresponding author. The data are not publicly available due to safeguarding employee privacy.

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
