# Peer review of "Epidemiological and Immunological Features of Obesity and SARS-CoV-2"

_viruses, 2021, doi:10.3390/v13112235_

Round 1

Reviewer 1 Report

Dear Editor, dear authors,

In their work, the authors Nilles EJ et al., attempt to study the very controversial subject of obesity and SARS-CoV2 infection. 

Although obesity is considered a negative prognostic marker for infection related survival in general, the exact pathophysiology driving this ineffectiveness to mount an adequate immune response remains elusive. 

Overall the experimental method(s) and results are nicely presented. I would suggest though that at least in the discussion section a more comprehensive approach towards the immunologic aspects of SARS-CoV2 infection as well as the immunology of obesity (only the highlights) should be attempted. A scheme may also help and make such a stiff and complicated subject more understandable namely, for example basic differences between healthy and obese immune response in SARS-CoV2 infection. Note that a description and distinction between the asymptomatic and the severe infection's immunology should be attempted. All these additions are, to my opinion, more than mandatory since the authors themselves measured several immunologic "players" in their work and present a theory involving the measured values. By explaining schematically and more comprehensively the above, their results shall be more solid and with much greater appeal.

Author Response

The authors would like to thank the reviewer for their time and thoughtful comments.

Reviewer 1

In their work, the authors Nilles EJ et al., attempt to study the very controversial subject of obesity and SARS-CoV2 infection. 

Although obesity is considered a negative prognostic marker for infection related survival in general, the exact pathophysiology driving this ineffectiveness to mount an adequate immune response remains elusive. 

Overall the experimental method(s) and results are nicely presented. I would suggest though that at least in the discussion section a more comprehensive approach towards the immunologic aspects of SARS-CoV2 infection as well as the immunology of obesity (only the highlights) should be attempted.

Response: Thank you for this comment. Additional description of the immunology of obesity has been added to the discussion.

A scheme may also help and make such a stiff and complicated subject more understandable namely, for example basic differences between healthy and obese immune response in SARS-CoV2 infection.

Response: Added Figure 6 that provides a schematic overview of the cytokines and inflammatory mediators that lead to dysfunction immune response among obese individuals.

Note that a description and distinction between the asymptomatic and the severe infection's immunology should be attempted.

Response: We agree that this is an important point. Unfortunately (or fortunately for the study participants), only a small number (n=5, 1.6%) of infections required hospitalizations and as such we are not powered to characterize differences in immune response by disease severity.

All these additions are, to my opinion, more than mandatory since the authors themselves measured several immunologic "players" in their work and present a theory involving the measured values. 

Response: Well received and the requested additions have been added.

By explaining schematically and more comprehensively the above, their results shall be more solid and with much greater appeal.

Response: Noted and added as suggested.

Reviewer 2 Report

The authors have studied the serological response to SARS-CoV-2 in a population of employees from an organization. About half of the employees volunteered in giving a blood sample and replying to a questionnaire about symptoms and baseline variables.

The study is well written and provides enough information for reviewing the quality of the research. The most striking finding is the difference in complaints in obese participants in the age group below 40 yrs of age.

The study shows interesting data and results. However, I have some comments.

The limitations in the discussion section is incomplete and should be extended. The study can be seriously biased by the limited response that is around 50% of the employees. The impact of this bias should be discussed. Is the group a representation of the complete population?

In addition, it is unclear if all patients produce enough antibodies to be detected in this study. Over time antibody titers may diminish. In that case the antibody negative group can include persons that have been infected. The low prevalence of antibodies (322/4469=7%) may indicate that this is true. Please address this issue in the discussion section.

Can the authors refer to publications concerning inflammatory responses in obese and non-obese patients because the premise is that symptoms are a result of inflammation? Are the current findings in line with this premise?

Author Response

The authors would like to thank the reviewer for their time and thoughtful comments.

Reviewer 2

The authors have studied the serological response to SARS-CoV-2 in a population of employees from an organization. About half of the employees volunteered in giving a blood sample and replying to a questionnaire about symptoms and baseline variables.

The study is well written and provides enough information for reviewing the quality of the research. The most striking finding is the difference in complaints in obese participants in the age group below 40 yrs of age.

Response: Thank you for your comment. We agree that the differences (and magnitude) in symptoms among obese individuals by age is very interesting and we hope leads to further targeted study of this phenomenon. 

The study shows interesting data and results. However, I have some comments.

The limitations in the discussion section is incomplete and should be extended. The study can be seriously biased by the limited response that is around 50% of the employees. The impact of this bias should be discussed. Is the group a representation of the complete population?

Response: Noted and additional limitations and implications added to the discussion section.

In addition, it is unclear if all patients produce enough antibodies to be detected in this study. Over time antibody titers may diminish. In that case the antibody negative group can include persons that have been infected. The low prevalence of antibodies (322/4469=7%) may indicate that this is true. Please address this issue in the discussion section.

Response: Noted and this limitation is added to discussion section.

Can the authors refer to publications concerning inflammatory responses in obese and non-obese patients because the premise is that symptoms are a result of inflammation? Are the current findings in line with this premise?

Response: Additional details on the inflammatory response in obese and non-obese individuals are included in the discussion section.

Round 2

Reviewer 1 Report

All comments have been taken into consideration. The manuscript has been substantially improved and is therefore, to my opinion eligible for publication. 

Reviewer 2 Report

The revisions have been made according to the comments that were made. I have no further comments.